# Information Directed Sampling for Sparse Linear Bandits

**Botao Hao**
Deepmind
haobotao000@gmail.com

**Tor Lattimore**
Deepmind
lattimore@google.com

**Wei Deng**
Department of Mathematics
Purdue University
weideng056@gmail.com

## Abstract

Stochastic sparse linear bandits offer a practical model for high-dimensional online decision-making problems and have a rich information-regret structure. In this work we explore the use of information-directed sampling (IDS), which naturally balances the information-regret trade-off. We develop a class of information-theoretic Bayesian regret bounds that nearly match existing lower bounds on a variety of problem instances, demonstrating the adaptivity of IDS. To efficiently implement sparse IDS, we propose an empirical Bayesian approach for sparse posterior sampling using a spike-and-slab Gaussian-Laplace prior. Numerical results demonstrate significant regret reductions by sparse IDS relative to several baselines.

## 1 Introduction

Standard linear bandits associate each action with a feature vector and assume the mean reward is the inner product between the feature vector and an unknown parameter vector [Auer, 2002, Dani et al., 2008, Rusmevichientong and Tsitsiklis, 2010, Chu et al., 2011, Abbasi-Yadkori et al., 2011]. Sparse linear bandits generalize linear bandits by assuming the unknown parameter vector is sparse [Abbasi-Yadkori et al., 2012, Carpentier and Munos, 2012, Hao et al., 2020b] and is of great practical significance for modeling high-dimensional online decision-making problems [Bastani and Bayati, 2020].

Lattimore and Szepesvári [2020, §24.3] established a $\Omega(\sqrt{sdn})$ regret lower bound for the *data-rich regime*, where $n$ is the horizon, $d$ is the feature dimension, $s$ is the sparsity and data-rich regime refers to the horizon $n \geq d^\alpha$ for some $\alpha > 0$. This means polynomial dependence on $d$ is generally not avoidable without additional assumptions. However, this bound hides much of the rich structure of sparse linear bandits by a crude maximisation over all environments.

When the action set admits a well-conditioned exploration distribution, Hao et al. [2020b] discovered the information-regret trade-off phenomenon by establishing an $\Theta(\text{poly}(s)n^{2/3})$ minimax rate for the *data-poor regime*. An interpretation for this optimal rate is that the agent needs to acquire enough information for fast sparse learning by pulling informative actions that even have high regret. Explore-then-commit algorithm can achieve this rate for the data-poor regime but is sub-optimal

in the data-rich regime. Therefore, our goal is to develop an efficient algorithm that can adapt to different information-regret structures for sparse linear bandits.

**Contributions**  Our contribution is three-fold:

- We prove that optimism-based algorithms fail to optimally address the information-regret trade-off in sparse linear bandits, which results in a sub-optimal regret bound.
- We provide the first analysis using information theory for sparse linear bandits and derive a class of nearly optimal Bayesian regret bounds for IDS that can adapt to information-regret structures.
- To approximate the information ratio, we develop an empirical Bayesian approach for sparse posterior sampling using spike-and-slab Gaussian-Laplace prior. Through several experiments, we justify the great empirical performance of sparse IDS with an efficient implementation.

## 2  Preliminary

We first introduce the basic setup of stochastic sparse linear bandits. The agent receives a compact action set $\mathcal{A} \subseteq \mathbb{R}^d$ in the beginning where $|\mathcal{A}| = K$. At each round $t$, the agent chooses an action $A_t \in \mathcal{A}$ and receives a reward $Y_t = \langle A_t, \theta^* \rangle + \eta_t$, where $(\eta_t)_{t=1}^n$ is a sequence of independent standard Gaussian random variables and $\theta^* \in \mathbb{R}^d$ is the true parameter unknown to the agent. We make the mild boundedness assumption that for all $a \in \mathcal{A}$, $\|a\|_\infty \leq 1$. The notion of sparsity can be defined through the parameter space $\Theta$:

$$\Theta = \left\{ \theta \in \mathbb{R}^d \,\middle|\, \sum_{j=1}^d \mathbb{1}\{\theta_j \neq 0\} \leq s, \|\theta\|_2 \leq 1 \right\}.$$

We assume $s$ is known and it can be relaxed by putting a prior on it. We consider the Bayesian setting where $\theta^*$ is a random variable taking values in $\Theta$ and denote $\rho$ as the prior distribution. The optimal action is $x^* = \operatorname{argmax}_{a \in \mathcal{A}} \mathbb{E}[\langle a, \theta^* \rangle | \theta^*]$. The agent chooses $A_t$ based on the history $\mathcal{F}_t = (A_1, Y_1, \ldots, A_{t-1}, Y_{t-1})$. Let $\mathcal{D}(\mathcal{A})$ be the space of probability measures over $\mathcal{A}$. A policy $\pi = (\pi_t)_{t \in \mathbb{N}}$ is a sequence of deterministic functions where $\pi_t(\mathcal{F}_t)$ specifies a probability distribution over $\mathcal{A}$. The information-theoretic Bayesian regret of a policy $\pi$ [Russo and Van Roy, 2014] is defined as

$$\mathfrak{BR}(n; \pi) = \mathbb{E}\left[ \sum_{t=1}^n \langle x^*, \theta^* \rangle - \sum_{t=1}^n Y_t \right],$$

where the expectation is over the interaction sequence induced by the agent and environment and the prior distribution over $\theta^*$.

**Notation**  Denote $I_d$ as the $d \times d$ identity matrix. Let $[n] = \{1, 2, \ldots, n\}$. For a vector $x$ and positive semidefinite matrix $A$, we let $\|x\|_A = \sqrt{x^\top A x}$ be the weighted $\ell_2$-norm and $\sigma_{\min}(A)$ be the minimum eigenvalue of $A$. The relation $x \gtrsim y$ means that $x$ is greater or equal to $y$ up to some universal constant and $\widetilde{O}(\cdot)$ hides modest logarithmic factors and universal constant. The cardinality of a set $\mathcal{A}$ is denoted by $|\mathcal{A}|$. Given a measure $\mathbb{P}$ and jointly distributed random variables $X$ and $Y$ we let $\mathbb{P}_X$ denote the law of $X$ and we let $\mathbb{P}_{X|Y}$ be the conditional law of $X$ given $Y$: $\mathbb{P}_{X|Y}(\cdot) = \mathbb{P}(X \in \cdot | Y)$. The mutual information between $X$ and $Y$ is $I(X; Y) = \mathbb{E}[D_{\mathrm{KL}}(\mathbb{P}_{X|Y} || \mathbb{P}_X)]$ where $D_{\mathrm{KL}}$ is the relative entropy. We write $\mathbb{P}_t(\cdot) = \mathbb{P}(\cdot | \mathcal{F}_t)$ as the posterior measure where $\mathbb{P}$ is the probability measure over $\theta$ and the history and $\mathbb{E}_t(\cdot) = \mathbb{E}(\cdot | \mathcal{F}_t)$. Denote $I_t(X; Y) = \mathbb{E}_t[D_{\mathrm{KL}}(\mathbb{P}_{t,X|Y} || \mathbb{P}_{t,X})]$.

## 3  Related work

**Sparse linear bandits**  Abbasi-Yadkori et al. [2012] proposed an inefficient online-to-confidence-set conversion approach that achieves an $\widetilde{O}(\sqrt{sdn})$ upper bound for an arbitrary action set. Lattimore

Table 1: Comparisons with existing results. APS11, LCS15, HLW20 refer to Abbasi-Yadkori et al. [2012], Lattimore et al. [2015], Hao et al. [2020b] accordingly. Exploratory action set is defined in Definition 5.2 and $K$ is the number of actions. The last lower bound is developed in Hao et al. [2020b].

| | Action set | Algorithm | Type | Rate |
|---|---|---|---|---|
| APS11 | arbitrary | online-to-confidence | freq | $\widetilde{O}(\sqrt{sdn})$ |
| LCS15 | hypercube | elimination | freq | $\widetilde{O}(s\sqrt{n})$ |
| HLW20 | exploratory | explore-then-commit | freq | $\widetilde{O}(s^{2/3}n^{2/3})$ |
| This paper | arbitrary | sparse IDS | Bayesian | $\widetilde{O}(\min(\sqrt{sdn}, \sqrt{dn\log(K)}))$ |
| This paper | arbitrary | sparse TS | Bayesian | $\widetilde{O}(\min(\sqrt{sdn}, \sqrt{dn\log(K)}))$ |
| This paper | exploratory | sparse IDS | Bayesian | $\widetilde{O}(\min(sn^{2/3}, \sqrt{sdn}))$ |
| Lower bound | arbitrary | NA | minimax | $\Omega(\sqrt{sdn})$ |
| Lower bound | exploratory | NA | minimax | $\Omega(\min(s^{1/3}n^{2/3}, \sqrt{dn}))$ |

et al. [2015] developed a selective explore-then-commit algorithm that only works when the action set is exactly the binary hypercube and derived an optimal $O(s\sqrt{n})$ upper bound. Hao et al. [2020b] introduced the notion of an exploratory action set and proved a $\Theta(\text{poly}(s)n^{2/3})$ minimax rate for the data-poor regime using an explore-then-commit algorithm. Hao et al. [2021] extended this concept to a MDP setting. Carpentier and Munos [2012] considered a special case where the action set is the unit sphere and the noise is vector-valued so that the noise becomes smaller as the dimension grows. We summarize the comparison of existing results with our work in Table 3.

**Sparse linear contextual bandits**  It recently became popular to study the contextual setting, where the action set changes from round to round. These results can not be reduced to our setting since they rely on either careful assumptions on the context distribution [Bastani and Bayati, 2020, Wang et al., 2018, Kim and Paik, 2019, Wang et al., 2020, Ren and Zhou, 2020, Oh et al., 2020] such that classical high-dimensional statistics can be used, or have polynomial dependency on the number of actions [Agarwal et al., 2014, Foster and Rakhlin, 2020, Simchi-Levi and Xu, 2020].

**Information-directed sampling**  Russo and Van Roy [2018] introduced IDS and derived Bayesian regret bounds for multi-armed bandits, linear bandits and combinatorial bandits. Liu et al. [2018] studied IDS for bandits with graph-feedback. Kirschner and Krause [2018], Kirschner et al. [2020a] investigated the use of IDS for bandits with heteroscedastic noise and partial monitoring. Kirschner et al. [2020b] proved the asymptotic optimality of frequentist IDS for linear bandits.

**Information-theoretic analysis**  Russo and Van Roy [2014] introduced an information-theoretic analysis of Thompson sampling (TS) and Bubeck and Sellke [2020] strengthened the result with a first-order Bayesian regret analysis. Dong and Roy [2018], Dong et al. [2019] extended the analysis to infinite-many actions and logistic bandits. Lattimore and Szepesvári [2019] explored the use of information-theoretic analysis for partial monitoring. Lu and Van Roy [2019] generalized the analysis to reinforcement learning.

**Bayesian sparse linear regression**  In the Bayesian framework, spike-and-slab methods are commonly used as probabilistic tools for sparse linear regression but most of prior works focus on variable selection and parameter estimation rather than uncertainty quantification [Mitchell and Beauchamp, 1988, George and McCulloch, 1993, Ročková and George, 2018]. Bai et al. [2020] provided a comprehensive overview.

## 4   Does the optimism optimally balance information and regret?

We demonstrate the necessity of balancing the trade-off between information and regret through a simple sparse linear bandit instance. We give an example where the optimal regret is only possible

by playing actions that are known to be sub-optimal. This phenomenon has disturbing implications for policies based on the principle of optimism, which is that they can never be minimax optimal in certain regime.

**Illustrative example**     Consider a problem instance where $\mathcal{A} = \mathcal{I} \cup \mathcal{U}$ is the union of an informative action set $\mathcal{I}$ and an uninformative action set $\mathcal{U}$:

- $\mathcal{I}$ is a *subset* of the hypercube that has three properties. First, $|\mathcal{I}| = O(s \log(ed/s))$. Second, the last coordinate of actions in $\mathcal{I}$ is always -1. Third, the empirical covariance of the uniform distribution over $\mathcal{I}$ has a restricted minimum eigenvalue (Definition A.1) at least 1/4. We prove such $\mathcal{I}$ does exist in Appendix A.1 through a probabilistic argument.

- $\mathcal{U} = \{x \in \mathbb{R}^d | x_j \in \{-1, 0, 1\} \text{ for } j \in [d-1], \|x\|_1 = s - 1, x_d = 0\}$.

The true parameter $\theta^* = (\varepsilon, \ldots, \varepsilon, 0, \ldots, 0, -1)$, where $\varepsilon > 0$ is a small constant.

**Information-regret structure**     Sampling an action uniformly at random from $\mathcal{I}$ ensures the covariance matrix is well-conditioned so that sparse learning such as Lasso [Tibshirani, 1996] can be used for learning $\theta^*$ faster than ordinary least squares. This means pulling actions from $\mathcal{I}$ provides more information to infer $\theta^*$ than from $\mathcal{U}$. On the other hand, actions from $\mathcal{I}$ lead to high regret due to the last coordinate -1. As a consequence, Hao et al. [2020b] has proven that the minimax regret for this problem is $\Theta(\text{poly}(s)n^{2/3})$ when the horizon is smaller than the ambient dimension.

**Sub-optimality of optimism-based algorithms**     We argue the optimism principle does not take this subtle trade-off into consideration and yields sub-optimal regret in the data-poor regime. In general, optimism-based algorithms choose $A_t = \text{argmax}_{a \in \mathcal{A}} \max_{\widetilde{\theta} \in \mathcal{C}_t} \langle a, \widetilde{\theta} \rangle$, where $\mathcal{C}_t \subseteq \mathbb{R}^d$ is a confidence set that contains the true $\theta^*$ with high probability. We assume that there exists a constant $c > 0$ such that $\mathcal{C}_t \subseteq \{\theta : (\widehat{\theta}_t - \theta)^\top V_t(\widehat{\theta}_t - \theta) \leq c\sqrt{s \log(n)}\}$, where $V_t = \sum_{s=1}^{t} A_s A_s^\top$ and $\widehat{\theta}_t$ is some sparsity-aware estimator. Such confidence set can be constructed through an online-to-confidence set conversion approach [Abbasi-Yadkori et al., 2012]. Define $\mathfrak{R}_\theta(n; \pi) = \mathbb{E}[\sum_{t=1}^{n} \langle x^*, \theta \rangle - \sum_{t=1}^{n} Y_t]$ for a fixed $\theta$.

**Claim 4.1.** Let $\pi^{\text{opt}}$ be such an optimism-based algorithm. There exists a sparse linear bandit instance characterized by $\theta$ such that for the data-poor regime, we have

$$\mathfrak{R}_\theta(n; \pi^{\text{opt}}) \gtrsim n/(\log(n)s\log(ed/s)).$$

The proof is deferred to Appendix A.1. The reason is that optimism-based algorithms do not choose actions for which they have collected enough statistics to prove these actions are suboptimal, but in the sparse linear setting it can be worth playing these actions when they are informative about other actions for which the statistics are not yet so clear. This phenomenon has been observed before in linear and structured bandits [Lattimore and Szepesvari, 2017, Combes et al., 2017, Hao et al., 2020a].

## 5 Information-directed sampling

As shown by Hao et al. [2020b], although the explore-then-commit algorithm can achieve the minimax optimal regret in the data-poor regime, it suffers sub-optimal regret in the data-rich regime. This motivates us consider IDS.

### 5.1 Design principle of IDS

Unlike the optimism principle, IDS explicitly balances the amount of information it gains about the optimal action and expected single-round regret through minimizing a notion of information ratio. More formally, when playing action $a$, the *information gain* $I_t(x^*; Y_{t,a})$ is the mutual information

between the optimal action and the reward the agent receives for taking action $a$, and the expected single-round regret is $\Delta_t(a) := \mathbb{E}_t[\langle x^*, \theta^* \rangle - \langle a, \theta^* \rangle]$. The information ratio is defined as

$$\Psi_{t,\lambda}(\pi) = \frac{(\Delta_t^\top \pi)^\lambda}{I_t^\top \pi} \, , \tag{5.1}$$

where we write $\Delta_t \in \mathbb{R}^{|\mathcal{A}|}$ and $I_t \in \mathbb{R}^{|\mathcal{A}|}$ as corresponding vectors. Then IDS takes the action according to $\pi_t = \operatorname{argmin}_{\pi \in \mathcal{D}(\mathcal{A})} \Psi_{t,2}(\pi)$.

**Remark 5.1.** The information ratio defined in Eq. (5.1) is a little more general than what Russo and Van Roy [2018] introduced which specified $\lambda = 2$. As observed by Lattimore and György [2020], the right value of $\lambda$ depends on the dependence of the regret on the horizon.

## 5.2 Information-theoretic Bayesian regret bound

In this section, we derive a class of Bayesian regret upper bound for sparse IDS. We first define a notion of exploratory action set.

**Definition 5.2** (Exploratory action set). Let $C_{\min}(\mathcal{A}) = \max_{\mu \in \mathcal{D}(\mathcal{A})} \sigma_{\min}(\mathbb{E}_{A \sim \mu}[AA^\top])$. For an action set $\mathcal{A}$, if $C_{\min}(\mathcal{A}) \geq 1$[1], we say $\mathcal{A}$ is exploratory.

We say that $\mathcal{A}$ has sparse optimal actions if the optimal action is $s$-sparse almost surely with respect to the prior. One can verify the action set of the hard instance developed in Hao et al. [2020b][2] is exploratory and has sparse optimal actions since sampling uniformly from the corner of informative action set shows that $C_{\min}(\mathcal{A}) \geq 1$ and the optimal actions always come from uninformative action set, which is sparse.

**Theorem 5.3** (Regret bound for sparse IDS). Suppose $\pi^{\text{IDS}} = (\pi_t)_{t \in \mathbb{N}}$ where $\pi_t = \operatorname{argmin}_\pi \Psi_{t,2}(\pi)$. Let $\Delta = \min(\log(K), 2s \log(Cdn^{1/2}/s))$ for some absolute constant $C > 0$. For an arbitrary action set, the following regret bound holds

$$\mathfrak{BR}(n; \pi^{\text{IDS}}) \leq \sqrt{\frac{1}{2} nd\Delta} \, .$$

When $\mathcal{A}$ is exploratory and has sparse optimal actions, the following regret bound holds

$$\mathfrak{BR}(n; \pi^{\text{IDS}}) \leq \min \left\{ \sqrt{\frac{1}{2} nd\Delta}, \frac{s^{\frac{2}{3}} n^{\frac{2}{3}} \Delta^{\frac{1}{3}}}{(2C_{\min}(\mathcal{A}))^{\frac{1}{3}}} \right\} \, .$$

This theorem shows the great adaptivity of IDS for sparse linear bandits in the sense that a single policy adapts to different information-regret structures. We summarize the regret bounds in a variety of of different regimes in Table 5.2.

**Remark 5.4.** The explore-then-commit algorithm proposed by Hao et al. [2020b] for sparse linear bandits has $O(\text{poly}(s)n^{2/3})$ regret bound when the action set is exploratory and it is known that this $O(n^{2/3})$ rate is not improvable. Thus, it is sub-optimal for data-rich regime comparing with $\Theta(\sqrt{sdn})$ minimax rate. In contrast, IDS is nearly optimal in both regimes.

As a direct application of our analysis, we also include a novel Bayesian regret bound for sparse TS.

**Corollary 5.5** (Regret bound for sparse TS). For an arbitrary action set, the following regret bound holds for some absolute constant $C > 0$

$$\mathfrak{BR}(n; \pi^{\text{TS}}) \leq \sqrt{\frac{1}{2} nd \min(\log(K), 2s \log(Cdn^{1/2}/s))} \, .$$

*Proof of Theorem 5.3.* We prove our main result in three steps. All the proofs of technical lemmas are deferred to the appendix.

---

[1] In the definition 1 is simply a convenient constant. More precisely, $C_{\min}(\mathcal{A})$ is a problem-dependent parameter of our regret bound.

[2] The hard instance is almost the same as the one in illustrative example except the informative action set is a full hypercube.

Table 2: Summary of regret bounds of IDS for different regimes. Data-rich regime refers to $n \gtrsim d^3\Delta/s^4$ and large $K$ refers to $K \gtrsim d\exp(s)$.

| | Arbitrary action set | Exploratory (data-rich) | Exploratory (data-poor) |
|---|---|---|---|
| Large $K$ | $\widetilde{O}(\sqrt{nds})$ | $\widetilde{O}(\sqrt{nds})$ | $\widetilde{O}(sn^{2/3})$ |
| Small $K$ | $\widetilde{O}(\sqrt{nd\log(K)})$ | $\widetilde{O}(\sqrt{nd\log(K)})$ | $\widetilde{O}(s^{2/3}n^{2/3}\log^{1/3}(K))$ |

**Step 1: Generic Bayesian regret upper bound**  We define $\Psi_{*,\lambda} \in \mathbb{R}$ as the *worse-case informa-tion ratio* such that for each $t \in [n]$, $\Psi_{t,\lambda}(\pi_t) \leq \Psi_{*,\lambda}$ almost surely.

**Lemma 5.6.** Suppose $\pi^{\text{IDS}} = (\pi_t)_{t\in\mathbb{N}}$ where $\pi_t = \arg\min_\pi \Psi_{t,2}(\pi)$. Then the following regret bound holds

$$\mathfrak{BR}(n;\pi^{\text{IDS}}) \leq \inf_{\lambda \geq 2} 2^{1-2/\lambda}\Psi_{*,\lambda}^{1/\lambda}I(x^*;\mathcal{F}_{n+1})^{1/\lambda}n^{1-1/\lambda},$$

where $\mathcal{F}_{n+1}$ refers to the history.

This lemma demonstrates the adaptivity of a single IDS for different information ratios. The choice of $\lambda$ essentially trades off the information-ratio and the horizon.

**Step 2: Bounding the worse-case information ratio**  We bound the worse-case information ratio for different $\lambda$. It shows that for certain action sets, the worse-case information ratio with $\lambda = 3$ could be much smaller than the one with $\lambda = 2$.

**Lemma 5.7.** For an arbitrary action set, we have $\Psi_{*,2} \leq d/2$. For an exploratory action set that has sparse optimal actions, we have $\Psi_{*,3} \leq s^2/(4C_{\min}(\mathcal{A}))$.

The bound of $\Psi_{*,2}$ essentially follows Russo and Van Roy [2014, Proposition 5] that bounds the information ratio of IDS by TS. And $\Psi_{*,3}$ is bounded by the information ratio of a mixture policy $\pi_t^{\text{mix}} = (1-\gamma)\pi_t^{\text{TS}} + \gamma\mu$ where $\mu$ is an exploratory policy such that $\sigma_{\min}(\mathbb{E}_{A\sim\mu}[AA^\top])$ is a constant and the mixture rate $\gamma \geq 0$ is optimized to minimize the bound.

**Step 3: Bounding the mutual information**  The mutual information $I(x^*;\mathcal{F}_{n+1})$ quantifies the cumulative information gain about the optimal action. Russo and Van Roy [2014, 2018] naively bound this term by entropy $H(x^*)$, which can be arbitrarily large or even infinite for some priors. Instead, we bound this term by the mutual information between the true parameter and the history through data-processing lemma.

**Lemma 5.8.** $I(x^*;\mathcal{F}_{n+1}) \leq I(\theta^*;\mathcal{F}_{n+1}) \leq \min\{\log(K), 2s\log(Cdn^{1/2}/s)\}$.

Our proof is based on the metric entropy of the parameter space and square root KL-divergence that is commonly used in information-theoretic lower bound analysis [Yang and Barron, 1999]. As a by-product of our analysis, by setting $s = d$, our analysis recovers the $\widetilde{O}(d\sqrt{n})$ Bayesian regret bound for TS under linear bandits with infinitely many actions without using rate-distortion theory [Dong and Roy, 2018]. Combining Lemmas 5.6-5.8 yields our conclusion. □

# 6 Computational methods

In this section, we provide an efficient implementation of sparse IDS. The main challenge is to generate posterior samples in a computationally efficient manner to approximate the information ratio. Due to the lack of conjugate prior, we propose an empirical Bayesian approach for sparse sampling with spike-and-slab priors.

## 6.1 An empirical Bayesian approach for sparse sampling

In the Bayesian framework, the golden standard for modeling $\theta^*$ is to place spike-and-slab priors [Mitchell and Beauchamp, 1988]. With a hierarchical structure over the parameter and model space,

vanilla spike-and-slab priors usually have the following form

$$\rho(\theta|\boldsymbol{\gamma}, \sigma^2) = \prod_{j=1}^{d} \left[\gamma_j \psi_1(\theta_j, \sigma) + (1 - \gamma_j)\psi_0(\theta_j, \sigma)\right], \rho(\boldsymbol{\gamma}|\beta) = \prod_{j=1}^{d} \beta^{\gamma_j}(1 - \beta)^{1-\gamma_j}, \qquad (6.1)$$

where $\boldsymbol{\gamma} = (\gamma_1, \ldots, \gamma_d)^\top$ is an intermediate binary vector that indexes the $2^d$ possible models and $\beta \in [0, 1]$ denotes a priori fraction of relevant variables among all the parameters. In particular, $\psi_1(\theta, \sigma)$ serves as a slab distribution to models relevant variables and $\psi_0(\theta, \sigma)$ is a point mass at zero that serves as a spike distribution to model irrelevant variables.

**Remark 6.1.** It is typical to assume $\beta$ follows a Beta prior as $\beta \sim \text{Beta}(a, b)$ and variance $\sigma^2$ follows an inverse gamma prior $\rho(\sigma^2) = \text{IG}(\nu/2, \nu\lambda/2)$ with $\nu = 1$ and $\lambda = 1$ (Rořková and George [2014]). For simplicity, we do not impose those additional layers of priors.

**Prior specification**. Although the prior in Eq. (6.1) is theoretically sound, exploring the full posterior in high-dimensions over the entire model space using point-mass spike-and-slab priors can be computationally prohibitive. Therefore, the spike distribution is usually relaxed as a small-scale Gaussian distribution [Rořková and George, 2014] or Laplace distribution [Ročková and George, 2018]. Thus, we consider a spike-and-slab Gaussian-Laplace prior that inherits the property of the Lasso [Tibshirani, 1996] for variable selection while the Gaussian component avoids the potential over-shrinkage effect. Each component of the prior is specified as

$$\psi_0(\theta, \sigma) = \frac{1}{2\sigma\lambda_0} \exp\left(-\frac{|\theta|}{\sigma\lambda_0}\right), \psi_1(\theta, \sigma) = \frac{1}{\sqrt{2\pi\sigma^2\lambda_1}} \exp\left(-\frac{\theta^2}{2\sigma^2\lambda_1}\right),$$

where $\lambda_0 > 0$ denotes a scaling parameter that encourages the shrinkage of irrelevant parameters and $\lambda_1$ is often set to a large value for a standard regularization [Rořková and George, 2014].

**An empirical Bayesian approach.** Suppose $\mathcal{L}(\mathcal{F}_{n+1}|\theta, \sigma^2)$ is the likelihood function where $Z_n$ is the historical data. According to the Bayes rule, the full posterior follows

$$p(\theta, \boldsymbol{\gamma}|\mathcal{F}_{n+1}, \sigma^2, \beta) \propto \mathcal{L}(\mathcal{F}_{n+1}|\theta, \sigma^2)\rho(\theta|\boldsymbol{\gamma}, \sigma^2)\rho(\boldsymbol{\gamma}|\beta). \qquad (6.2)$$

To speed up the sampling, we only sample $\theta$ and optimize $\boldsymbol{\gamma}$ with respect to the posterior instead. To tackle this issue of the binary vector $\boldsymbol{\gamma}$, we consider a continuous relaxation by introducing a latent vector $\nu \in [0, 1]^d$ as the probability of the variable being included in the model. We focus on the simulations of the conditional expectation of the complete posterior:

$$\mathbb{E}_{\boldsymbol{\gamma}|\cdot} \left[\log p(\theta, \boldsymbol{\gamma}|\mathcal{F}_{n+1}, \sigma^2, \beta)\right] = \log \mathcal{L}(\mathcal{F}_{n+1}|\theta, \sigma^2) + \mathbb{E}_{\boldsymbol{\gamma}|\cdot} \left[\log \rho(\boldsymbol{\gamma}|\beta) + \log \rho(\theta|\boldsymbol{\gamma}, \sigma^2)\right] + C_1,$$

where $C_1$ is the normalizing constant and $\mathbb{E}_{\boldsymbol{\gamma}|\cdot}[\cdot]$ denotes the the conditional expectation with respect to $\gamma$ given the current parameter $\theta$. Then we compute

$$\mathbb{E}_{\boldsymbol{\gamma}|\cdot} [\log \rho(\boldsymbol{\gamma}|\beta)] = \sum_{i=1}^{d} \mathbb{E}_{\boldsymbol{\gamma}|\cdot} [\gamma_i \log(\beta) + (1 - \gamma_i)\log(1 - \beta)] = \sum_{i=1}^{d} \log\left(\frac{\beta}{1 - \beta}\right)\nu_i + C_2,$$

where $C_2 = d\log(1 - \beta)$ and $\nu_i = \mathbb{E}_{\boldsymbol{\gamma}|\cdot}[\gamma_i]$ denotes a conditional probability. By the Bayes rule, we have

$$\nu_i = \mathbb{P}(\gamma_i = 1|\theta_i, \beta) = \frac{\rho(\theta_i|\gamma_i = 1)\mathbb{P}(\gamma_i = 1|\beta)}{\rho(\theta_i|\gamma_i = 1)\mathbb{P}(\gamma_i = 1|\beta) + \rho(\theta_i|\gamma_i = 0)\mathbb{P}(\gamma_i = 0|\beta)}. \qquad (6.3)$$

For the mixture prior, we optimize the variational lower bound

$$\mathbb{E}_{\boldsymbol{\gamma}|\cdot}[\log \rho(\theta|\boldsymbol{\gamma}, \sigma^2)] \geq \sum_{j=1}^{d} -\frac{1 - \nu_j}{\lambda_0} \frac{|\theta_j|}{\sigma} - \frac{\nu_j}{\lambda_1} \frac{\theta_j^2}{2\sigma^2} + C_3,$$

where the inequality follows by Jensen's inequality and $C_3$ denotes a trivial constant.

We summarize the full sampling procedure in Algorithm 1. Given a current estimate of $(\theta^{(k)}, \nu^{(k)})$ at step $k$ and $\mathcal{F}_{n+1}$, we adapt an empirical Bayesian method [Deng et al., 2019] by iteratively sampling $\theta$ based on the negative log-posterior with adaptive priors

$$Q(\theta|\theta^{(k)}, \nu^{(k)}, \mathcal{F}_{n+1}) = -\log \mathcal{L}(\mathcal{F}_{n+1}|\theta, \sigma^2) + \sum_{i=1}^{d} \left(\frac{1 - \rho_i}{\lambda_0} \frac{|\theta_i|}{\sigma} + \frac{\rho_i}{\lambda_1} \frac{\theta_i^2}{2\sigma^2}\right),$$

and optimizing the conditional probability $\nu$ through stochastic approximation algorithms (Robbins and Monro [1951]) until the equilibrium is achieved.

**Remark 6.2.** Algorithm 1 will need to output $M$ posterior samples. Suppose we run a Markov Chain for 100 steps ($k = 1, 2, \ldots, 100$) and want 10 posterior samples. Setting the thinning factor $T = 10$ means every 10 steps, we pick a $\theta^k$ as the required posterior sample, e.g., $\theta^{10}, \theta^{20}, \ldots, \theta^{100}$.

---

**Algorithm 1** Empirical Bayesian sparse sampling

1: **Input:** dataset $\mathcal{F}_{n+1}$, learning rate $(\eta_k)$, step size $(\omega_k)$, a priori knowledge of $\sigma^2$ and $\beta$, thinning factor $T$, number of posterior samples $M$, regularization parameters $\lambda_0, \lambda_1$.
2: **Initialize:** $\theta^{(0)} \sim N(0, 0.1I_d)$ and $\nu^{(0)} = (0.5, \ldots, 0.5)^\top$.
3: **for** $k \geq 1$ **do**
4:     Sampling step: $\theta^{(k+1)} = \theta^{(k)} - \eta_k \frac{\partial}{\partial \theta} Q(\theta|\theta^{(k)}, \nu^{(k)}, \mathcal{F}_{n+1}) + \sqrt{2\eta_k}\xi^{(k)}$, where $\xi^{(k)}$ is a standard Gaussian random vector.
5:     Stochastic approximation step: $\nu^{(k+1)} = (1 - \omega_k)\nu^{(k)} + \omega_k \nu$, where $\nu$ is derived from Eq. (6.3).
6: **end for**
7: **Output:** $M$ posterior samples $\theta^{(T)}, \theta^{(2T)}, \ldots, \theta^{(MT)}$.

---

## 6.2 Optimize the information ratio

For sparse linear bandits, it is expensive to estimate and optimize the original information ratio that involves the calculation of KL-divergence. Following Section 6.3 in Russo and Van Roy [2018], we optimize a variance-based information ratio instead: $\pi_t = \operatorname{argmin}_\pi (\Delta_t^\top \pi)^2/(2v_t^\top \pi)$, where we define $v_t(a) = \mathbb{E}_t[a^\top \mathbb{E}_t[\theta^*|x^*] - a^\top \mathbb{E}_t[\theta^*]]^2$ for each $a \in \mathcal{A}$. With sufficient number of posterior samples of $\theta^*$ produced by Algorithm 1, we can accurately estimate $\mathbb{E}_t[\theta^*|x^*], \mathbb{E}_t[\theta^*]$ and the information ratio. The detailed procedure is deferred to Appendix B. For the optimization step, we simply choose the action who can has the minimum per-action variance-based information ratio to accelerate the computation. The overall algorithm can be found in Algorithm 2.

---

**Algorithm 2** Sparse IDS

1: **Input:** time horizon $n$, action set $\mathcal{A}$, number of posterior samples per round $M$.
2: **for** $t = 1, \cdots, n$ **do**
3:     Obtain $M$ posterior samples $\theta^1, \ldots, \theta^M$ by Algorithm 1.
4:     Calculate $\widehat{\Delta}_t \in \mathbb{R}^{|\mathcal{A}|}$ and $\widehat{v}_t \in \mathbb{R}^{|\mathcal{A}|}$ using Algorithm 3 in Appendix B.
5:     Take the action $A_t = \operatorname{argmin}_{a \in \mathcal{A}} \widehat{\Delta}_t^2(a)/\widehat{v}_t(a)$ and receive a reward: $Y_t = \langle A_t, \theta^* \rangle + \eta_t$.
6: **end for**

---

**Remark 6.3.** While sparse IDS is computationally efficient, we only have samples from approximate posterior distribution rather than the exact one. This will incur a non-negligible approximation error that may depend linearly on the time horizon. This is observed previously by analyzing approximate Thompson sampling [Lu and Van Roy, 2017] or more recently for IDS [Lu et al., 2021].

## 7 Experiments

First, we evaluate the performance of the empirical Bayesian sparse sampling procedure for generating posterior samples through an offline sparse linear regression. We set $d = 10, s = 3, n = 100$ and the actions are drawn i.i.d from a multivariate normal distribution $N(0, \Sigma)$ with $\Sigma_{ij} = 0.6^{|i-j|}$. The true parameter is $\theta^* = (3, 2, 0, 0, \ldots, 0) \in \mathbb{R}^{10}$. We plot the empirical posterior distributions as well as their posterior mean for the first three covariates in Figure 1. It shows that the posterior distribution concentrates well around the true value and the algorithm identifies the sparse pattern quickly.

Second, we evaluate sparse IDS with several other competitors. In particular, we compare with LinUCB [Abbasi-Yadkori et al., 2011], LinTS with Gaussian prior [Agrawal and Goyal, 2013], IDS

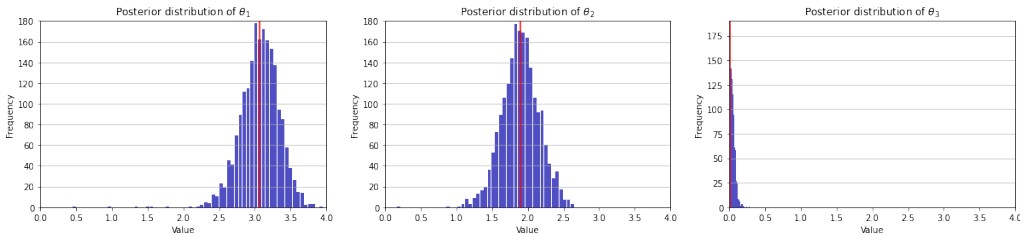

Figure 1: Posterior distributions for the first three coordinates. The red lines are posterior means.

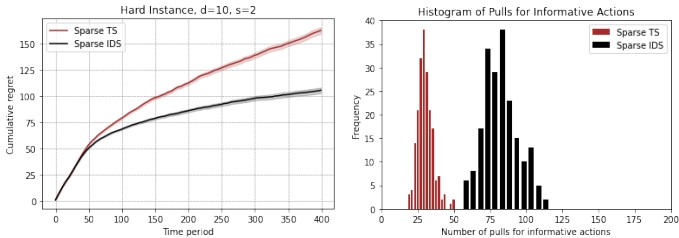

Figure 2: The left panel is the cumulative regret and the right panel is the histogram of number of pulls for informative actions. It's clear that sparse TS does not value information enough as it should.

for linear bandits (Algorithm 6 in Russo and Van Roy [2018]) and ESTC [Hao et al., 2020b]. Note that the first three algorithms are not sparsity-aware. Our sparse sampling procedure naturally induces a sparse TS algorithm so we include it into comparison.

**Setting** All the true parameters are randomly generated from a multivariate normal distribution, truncated to be sparse and normalized to have square norm 1. The noise variance is fixed to be 2 and we replicate the experiments over 200 trials. We plot the empirical cumulative Bayesian regret. Each Bayesian algorithm will take 10000 posterior samples. We use the TS without blow-up factor for the variance and tune the length of confidence interval of LinUCB over a candidate set.

**Hard sparse linear bandits instance** Consider the hard problem instance introduced in Section 4 that includes informative and uninformative action sets and set $d = 10, s = 2$. For each trial, we record the number of pulls of sparse TS and sparse IDS for informative actions. We draw the histogram of number of pulls during 200 trial in Figure 2. It is clear that IDS tends to invest more on the informative action sets but suffer less regret than TS if there exists an information-regret trade-off phenomenon.

**Multivariate Gaussian action set** We consider a more general case where each action is generated from multivariate normal distribution $N(0, \Sigma)$ with $\Sigma_{ij} = 0.6^{|i-j|}$. The number of actions $K$ is fixed to be 200 and the level of sparsity $s/d$ is fixed to be 0.1. We report the results in Figure 3 for $d = 20, 40, 100$. It is obvious that sparse IDS consistently outperforms other algorithms and the improvement increases as the feature dimension increases.

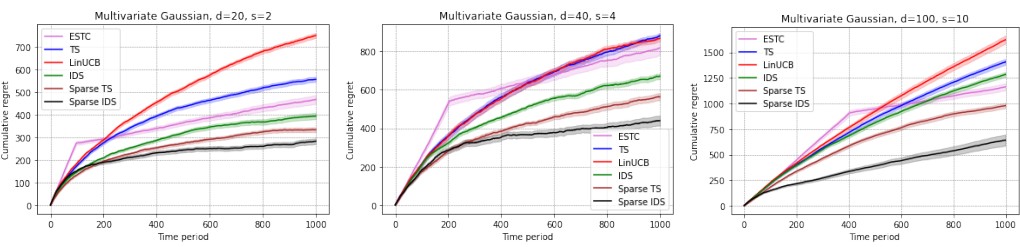

Figure 3: Cumulative regret for $d = 20, 40, 100$.

## 8 Conclusion

In this work, we investigate the theoretic and practical applicability of information-directed sampling for sparse linear bandits. An interesting future direction is to extend similar ideas to sparse linear contextual bandits.

## Acknowledgement

We thank Yoan Russac, Dorian Baudry and Alexandre Filiot for making their code of information-directed sampling public.

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
