# A  Proofs

## A.1  Proof of Claim 4.1

We first define the notion of restricted minimum eigenvalue.

**Definition A.1** (Restricted minimum eigenvalue). Given a symmetric matrix $H \in \mathbb{R}^{d \times d}$ and integer $s \geq 1$, and $L > 0$, the restricted minimum eigenvalue of $H$ is defined as

$$\phi^2(H, s, L) := \min_{\mathcal{S} \subset [d], |\mathcal{S}| \leq s} \min_{\theta \in \mathbb{R}^d} \left\{ \frac{\langle \theta, H\theta \rangle}{\|\theta_{\mathcal{S}}\|_2^2} : \theta \in \mathbb{R}^d, \|\theta_{\mathcal{S}^c}\|_1 \leq L \|\theta_{\mathcal{S}}\|_1 \right\}.$$

Suppose $\{x^{(t)}\}_{t=1}^k \subseteq \mathbb{R}^d$ are $k$ independent random vectors who first $d-1$ coordinates are drawn uniformly from $\{-1, 1\}$ and the last coordinate is 1. Denote $\widehat{\Sigma} = \sum_{t=1}^k x^{(t)} x^{(t)\top}$. It is easy to see $\mathbb{E}[\widehat{\Sigma}] = I_d$ and $\sigma_{\min}(\mathbb{E}[\widehat{\Sigma}]) = 1$. From the definition of restricted minimum eigenvalue, we have for any $L > 0$,

$$\phi^2(\mathbb{E}[\widehat{\Sigma}], s, L) \geq \sigma_{\min}^2(\mathbb{E}[\widehat{\Sigma}]) = 1.$$

According to Theorem 10 in Javanmard and Montanari [2014] (essentially from Theorem 6 in Rudelson and Zhou [2013]), if the population covariance matrix satisfies the restricted eigenvalue condition, the empirical covariance matrix satisfies it as well with high probability. Specifically, when $k = C_1 s \log(ed/s)$ for some large constant $C_1 > 0$, the following holds:

$$\mathbb{P}\left( \phi^2(\widehat{\Sigma}, s, 3) \geq \frac{1}{4} \right) \geq 1 - 2\exp(-k/C_1) \geq 0.5.$$

According to probabilistic argument, there exists a set of fixed actions $\{x^{(1)}, \ldots, x^{(k)}\}$ with $k = C_1 s \log(ed/s)$ such that if we pull uniformly at random from them, the restricted minimum eigenvalue of the resulting covariance matrix is at least $1/4$.

Next we compute how many rounds at most the optimism-based algorithm will choose from informative action set $\mathcal{I}$. Let $N_{t-1}(a)$ as the number of pulls for action $a$ until round $t$. Since we have $\theta^* \in \mathcal{C}_t$ with high probability, then

$$\max_{a \in \mathcal{U}} \max_{\widetilde{\theta} \in \mathcal{C}_t} \langle a, \widetilde{\theta} \rangle \geq \max_{a \in \mathcal{U}} \langle a, \theta^* \rangle \geq s\varepsilon.$$

On the other hand for any action $a \in \mathcal{I}$,

$$\max_{\widetilde{\theta} \in \mathcal{C}_t} \langle a, \widetilde{\theta} \rangle = \max_{\widetilde{\theta} \in \mathcal{C}_t} \langle a, \widetilde{\theta} - \theta^* \rangle + \langle a, \theta^* \rangle \leq \max_{\widetilde{\theta} \in \mathcal{C}_t} \langle a, \widetilde{\theta} - \theta^* \rangle + \max_{a \in \mathcal{I}} \langle a, \theta^* \rangle$$

$$= \max_{\widetilde{\theta} \in \mathcal{C}_t} \langle a, \widetilde{\theta} - \theta^* \rangle + s\varepsilon - 1 \leq 2c\sqrt{\|a\|_{V_t^{-1}} s \log(n)} + s\varepsilon - 1$$

$$\leq 2c\sqrt{\frac{s \log(n)}{N_{t-1}(a)}} + s\varepsilon - 1.$$

If $N_{t-1}(a) > 4c^2 s \log(n)$ for $a \in \mathcal{I}$, then we have $\max_{\widetilde{\theta} \in \mathcal{C}_t} \langle a, \widetilde{\theta} \rangle < s\varepsilon$. Based on the optimism principle, the algorithm will switch to pull uninformative actions. This leads to the fact that optimism-based algorithm will pull at most $|\mathcal{I}| 4c^2 s \log(n)$ rounds of information actions. According to the proof of minimax lower bound in Hao et al. [2020b], we have when $\sum_{a \in \mathcal{I}} N_n(a) < 1/(s\varepsilon^2)$, there exists another sparse parameter $\theta'$ such that

$$R_\theta(n) + R_{\theta'}(n) \gtrsim ns\varepsilon \exp\left( -\frac{2n\varepsilon^2 s^2}{d} \right). \tag{A.1}$$

By choosing $\varepsilon = \sqrt{1/(s^2 \log(n) 4c^2 |\mathcal{I}|)}$, we have for $d \geq n/(s \log(n) \log(ed/s))$

$$R_\theta(n) + R_{\theta'}(n) \gtrsim \frac{n}{\sqrt{\log(n)|\mathcal{I}|}}. \tag{A.2}$$

Note that $|\mathcal{I}| = O(s \log(d/s))$ as we proved before. Then we can argue there exists a sparse linear bandit instance such that optimism-based algorithm will suffer linear regret for a data-poor regime. This ends the proof.

## A.2 Proof of Lemma 5.6

We decompose the Bayesian regret in terms of the instantaneous regret:

$$\mathfrak{BR}(n; \pi^{\text{IDS}}) = \mathbb{E}\left[\sum_{t=1}^{n}\langle x^*, \theta\rangle - \sum_{t=1}^{n} Y_t\right] = \mathbb{E}\left[\sum_{t=1}^{n}\mathbb{E}_t\left[\langle x^*, \theta^*\rangle - Y_t\right]\right]$$

$$= \mathbb{E}\left[\sum_{t=1}^{n}\sum_a \mathbb{E}_t\left[\langle x^*, \theta^*\rangle - \langle a, \theta^*\rangle\right]\pi_t(a)\right] = \mathbb{E}\left[\sum_{t=1}^{n}\langle \pi_t, \Delta_t\rangle\right], \tag{A.3}$$

where the third equation is due to the zero mean of the noise.

We then bound one-step instantaneous regret. From the definition of $\pi_t$, we have

$$\pi_t = \operatorname*{argmin}_{\pi \in \mathcal{D}(\mathcal{A})} \frac{\langle \pi, \Delta_t\rangle^2}{\langle \pi, I_t\rangle}. \tag{A.4}$$

In addition, we denote

$$q_{\lambda,t} = \operatorname*{argmin}_{\pi \in \mathcal{D}(\mathcal{A})} \Psi_{t,\lambda}(\pi) = \operatorname*{argmin}_{\pi \in \mathcal{D}(\mathcal{A})} \frac{\langle \pi, \Delta_t\rangle^\lambda}{\langle \pi, I_t\rangle}. \tag{A.5}$$

Note that

$$\nabla_\pi \Psi_{t,2}(\pi) = \frac{2\langle \pi, \Delta_t\rangle\Delta_t}{\langle \pi, I_t\rangle} + \frac{\langle \pi, \Delta_t\rangle^2 I_t}{\langle \pi, I_t\rangle^2}.$$

By the first-order optimality condition in Lemma C.1,

$$0 \le \langle \nabla_\pi \Psi_{t,2}(\pi_t), q_{\lambda,t} - \pi_t\rangle = \frac{2\langle q_{\lambda,t} - \pi_t, \Delta_t\rangle\langle \pi_t, \Delta_t\rangle}{\langle \pi_t, I_t\rangle} - \frac{\langle q_{\lambda,t} - \pi_t, I_t\rangle\langle \pi_t, \Delta_t\rangle^2}{\langle \pi_t, I_t\rangle^2}.$$

This further implies

$$2\langle q_{\lambda,t}, \Delta_t\rangle \ge \langle \pi_t, \Delta_t\rangle\left(1 + \frac{\langle q_{\lambda,t}, I_t\rangle}{\langle \pi_t, I_t\rangle}\right) \ge \langle \pi_t, \Delta_t\rangle.$$

Based on the above equation, we can bound the generalized information ratio as follows:

$$\frac{\langle \pi_t, \Delta_t\rangle^\lambda}{\langle \pi_t, I_t\rangle} = \frac{\langle \pi_t, \Delta_t\rangle^2\langle \pi_t, \Delta_t\rangle^{\lambda-2}}{\langle \pi_t, I_t\rangle} \le \frac{2^{\lambda-2}\langle \pi_t, \Delta_t\rangle^2\langle q_{\lambda,t}, \Delta_t\rangle^{\lambda-2}}{\langle \pi_t, I_t\rangle}$$

$$\le \frac{2^{\lambda-2}\langle q_{\lambda,t}, \Delta_t\rangle^{\lambda-2}\langle q_{\lambda,t}, \Delta_t\rangle^2}{\langle q_{\lambda,t}, I_t\rangle} = 2^{\lambda-2}\min_{\pi \in \mathcal{D}(\mathcal{A})}\frac{\langle \pi, \Delta_t\rangle^\lambda}{\langle \pi, I_t\rangle},$$

where the first inequality is from Eq. (A.4) and the second inequality is from Eq. (A.5). According to the definition of $\Psi_{*,\lambda}$, we have

$$\langle \pi_t, \Delta_t\rangle \le 2^{1-2/\lambda}\langle \pi_t, I_t\rangle^{1/\lambda}\Psi_{*,\lambda}^{1/\lambda}.$$

Next we prove $\langle \pi_t, I_t\rangle = I_t(x^*; (A_t, Y_t))$. By the chain rule of mutual information,

$$I_t(x^*; (A_t, Y_t)) = I_t(x^*; A_t) + \mathbb{E}_t[I_t(x^*; Y_t|A_t)] = \mathbb{E}_t[I_t(x^*; Y_t|A_t)]$$

$$= \sum_{a \in \mathcal{A}}\pi_t(a)I_t(x^*; Y_t|A_t = a),$$

where we use the fact that $A_t$ and $x^*$ are independent. If $Z$ is independent of $X$ and $Y$, then we have $I(X; Y|Z) = I(X; Y)$. Since $A_t$ is independent of $x^*$ and $Y_t$ conditional on $\mathcal{F}_t$, then

$$\sum_{a \in \mathcal{A}}\pi_t(a)I_t(x^*; Y_t|A_t = a) = \sum_{a \in \mathcal{A}}\pi_t(a)I_t(x^*; Y_{t,a}) = \langle \pi_t, I_t\rangle.$$

This proves the previous claim. Combining with Eq. (A.3),

$$\mathfrak{BR}(n; \pi^{\text{IDS}}) = \mathbb{E}\left[\sum_{t=1}^n \langle \pi_t, \Delta_t \rangle\right] \leq \mathbb{E}\left[\sum_{t=1}^n 2^{1-2/\lambda} I_t(x^*; (A_t, Y_t))^{1/\lambda} \Psi_{*,\lambda}^{1/\lambda}\right]$$

$$= 2^{1-2/\lambda} \Psi_{*,\lambda}^{1/\lambda} \mathbb{E}\left[\sum_{t=1}^n I_t(x^*; (A_t, Y_t))^{1/\lambda}\right] \tag{A.6}$$

$$\leq 2^{1-2/\lambda} \Psi_{*,\lambda}^{1/\lambda} n^{1-1/\lambda} \mathbb{E}\left[\sum_{t=1}^n I_t(x^*; (A_t, Y_t))\right]^{1/\lambda},$$

where the last inequality is from Holder's inequality with $p = \lambda/(\lambda - 1)$ and $q = \lambda$.

In the end, we bound the cumulative information gain using the chain rule of mutual information,

$$\sum_{t=1}^n \mathbb{E}[I_t(x^*; (A_t, Y_t))] = \sum_{t=1}^n I(x^*; (A_t, Y_t)|\mathcal{F}_t) = I(x^*; \mathcal{F}_{n+1}).$$

Combining with Eq. (A.6), we have

$$\mathfrak{BR}_n(\pi, \rho) \leq 2^{1-2/\lambda} (\Psi_{*,\lambda} I(x^*; \mathcal{F}_{n+1}))^{1/\lambda} n^{1-1/\lambda}.$$

This ends the proof.

## A.3 Proof of Lemma 5.8

Denote $Z_1 = (A_1, Y_1), \ldots, Z_n = (A_n, Y_n)$ such that $Z^n = (Z_1, \ldots, Z_n)$. When the number of actions $K$ is small, we could directly bound it by

$$I(x^*; Z^n) = H(x^*) - H(x^*|Z^n) \leq H(x^*) \leq \log|\mathcal{A}| = \log(K),$$

where for the first inequality we use the non-negativity of Shannon entropy.

When the number of actions is large or infinite, we will bound it through the following information-theoretic argument. Recall that $x^* = \operatorname{argmin}_{a \in \mathcal{A}} x^\top \theta^*$ so $x^*$ can be viewed as a deterministic function $\theta^*$. By the data processing lemma (Lemma C.2), we have $I(x^*; Z^n) \leq I(\theta^*; Z^n)$. In other words, we bound the information gain regarding the optimal action by the information gain regarding the true parameter.

Recall that we assume the prior distribution of $\theta^*$ is $\rho(\theta^*)$ that takes the value in $\Theta$. From Vershynin [2009] , we know $\Theta$ enjoys an $\varepsilon$-net $\mathcal{N}_\varepsilon$ under $\ell_2$-norm and its cardinality at most $(Cd/s\varepsilon)^s$ where $C$ is a constant. Hence, its metric entropy satisfies

$$\log|\mathcal{N}_\varepsilon| \leq s \log(Cd/s\varepsilon). \tag{A.7}$$

Suppose the Bayes mixture density $p_\rho(z^n) = \int_{\theta \in \Theta} p(z^n|\theta) d\rho(\theta)$. According to the definition of mutual information,

$$I(\theta^*; Z^n) = \mathbb{E}_{\theta^*}\left[D_{\text{KL}}(\mathbb{P}_{Z^n|\theta^*}||\mathbb{P}_{Z^n})\right]$$

$$= \int_{\theta^* \in \Theta} \int p(z^n|\theta^*) \log\left(\frac{p(z^n|\theta^*)}{p_w(z^n)}\right) \mu(dz^n) d\rho(\theta^*)$$

$$\leq \int_{\theta \in \Theta} \int p(z^n|\theta^*) \log\left(\frac{p(z^n|\theta^*)}{q(z^n)}\right) \mu(dz^n) d\rho(\theta^*) \tag{A.8}$$

$$= \int_{\theta \in \Theta} D_{\text{KL}}(\mathbb{P}_{Z^n|\theta^*}||\mathbb{Q}_{Z^n}) d\rho(\theta^*).$$

where the inequality is due to the fact that Bayes mixture density $p_\rho(z^n)$ minimizes the average KL divergences over any choice of densities $q(z^n)$. Then we choose $\rho_1$ as an uniform distribution over $\mathcal{N}_\varepsilon$ such that $q(z^n) = p_{\rho_1}(z^n) = \int_{\theta \in \Theta} p(z^n|\theta) d\rho_1(\theta)$ and we denote $\mathbb{Q}_{Z^n}$ as the corresponding

probability measure. Since $\mathcal{N}_\varepsilon$ is an $\varepsilon$-net over $\Theta$ under $\ell_2$-norm, for each $\theta \in \Theta$, there exists $\widetilde{\theta} \in \Theta$ such that $\|\theta - \widetilde{\theta}\|_2 \leq \varepsilon$.

To bound the KL-divergence term, we follow

$$
\begin{aligned}
D_{\mathrm{KL}}(\mathbb{P}_{Z^n|\theta}||\mathbb{Q}_{Z^n}) &= \mathbb{E}\left[\log \frac{p(z^n|\theta^*)}{(1/|\mathcal{N}_\varepsilon|)\sum_{\widetilde{\theta}\in\mathcal{N}_\varepsilon} p(z^n|\widetilde{\theta})}\right] \\
&\leq \mathbb{E}\left[\log \frac{p(z^n|\theta^*)}{(1/|\mathcal{N}_\varepsilon|)p(z^n|\widetilde{\theta})}\right] \\
&\leq \log|\mathcal{N}_\varepsilon| + D_{\mathrm{KL}}(\mathbb{P}_{Z^n|\theta}||\mathbb{P}_{Z^n|\widetilde{\theta}}).
\end{aligned}
\tag{A.9}
$$

By the chain rule of KL-divergence,

$$
D_{\mathrm{KL}}(P_{Z^n|\theta}||\mathbb{P}_{Z^n|\widetilde{\theta}}) \leq \mathbb{E}\left[\sum_{t=1}^n D_{\mathrm{KL}}(\mathbb{P}_{Y_t|A_t,Z^{t-1},\theta^*}||\mathbb{P}_{Y_t|A_t,Z^{t-1},\widetilde{\theta}})\right],
$$

where we define $Z^0 = \emptyset$. Under linear model and bandit $\theta$, we know $Y_t \sim N(A_t^\top\theta, 1)$. A straightforward computation leads to

$$
\begin{aligned}
D_{\mathrm{KL}}(\mathbb{P}_{Y_t|A_t,Z^{t-1},\theta^*}||\mathbb{P}_{Y_t|A_t,Z^{t-1},\widetilde{\theta}}) &= \frac{1}{2\sigma^2}\|A_t^\top\theta^* - A_t^\top\widetilde{\theta}\|_2^2 \\
&\leq \frac{1}{2\sigma^2}\|A_t\|_\infty^2\|\theta^* - \widetilde{\theta}\|_1^2 \\
&\leq \frac{1}{2\sigma^2}s\|\theta^* - \widetilde{\theta}\|_2^2 \\
&\leq \frac{s}{2\sigma^2}\varepsilon^2,
\end{aligned}
\tag{A.10}
$$

where the first inequality we use the fact that $\|a\|_\infty \leq 1$ and the parameters are sparse. Here actually we only require $\|a\|_\infty$ for $a \in \mathcal{A}$ being bounded by a constant since evetually it will only appears inside the logarithm term. Putting Eqs. (A.7)-(A.10) together, we have

$$
I(\theta^*; Z^n) \leq \int_{\theta^*\in\Theta}\left(s\log(Cd/s\varepsilon) + \frac{ns}{2\sigma^2}\varepsilon^2\right)d\theta^* = s\log(Cd/s\varepsilon) + \frac{ns}{2\sigma^2}\varepsilon^2.
$$

With the choice of $\varepsilon = 1/\sqrt{n}$, we finally have

$$
I(\theta^*; Z^n) \leq 2s\log(Cdn^{1/2}/s).
$$

This ends the proof.

## A.4 Proof of Lemma 5.7

For any particular policy $\widetilde{\pi}$, if one can derive an worse-case bound of $\Psi_{t,\lambda}(\widetilde{\pi})$, we get an upper bound for $\Psi_{*,\lambda}$ automatically. The remaining step is to choose proper policy $\widetilde{\pi}$.

First, we bound the information ratio with $\lambda = 2$ that essentially follows Proposition 5 in Russo and Van Roy [2014] and Lemma 3 in Russo and Van Roy [2014] for a Gaussian noise. By the definition of mutual information, for any $a \in \mathcal{A}$, we have

$$
\begin{aligned}
I_t(x^*; Y_{t,a}) &= D_{\mathrm{KL}}\left(\mathbb{P}_t((x^*, Y_{t,a}))||\mathbb{P}_t(x^* \in \cdot)\mathbb{P}_t(Y_{t,a} \in )\right) \\
&= \sum_{a^*\in\mathcal{A}}\mathbb{P}_t(x^* = a^*)D_{\mathrm{KL}}\left(\mathbb{P}_t(Y_{t,a} = \cdot|x^* = a^*)||\mathbb{P}_t(Y_{t,a} = \cdot)\right).
\end{aligned}
\tag{A.11}
$$

Define $R_{\max}$ as the upper bound of maximum expected reward. It is easy to see $Y_{t,a}$ is a $\sqrt{R_{\max}^2 + 1}$ sub-Gaussian random variable. According to Lemma 3 in Russo and Van Roy [2014], we have

$$
I_t(x^*; Y_{t,a}) \geq \frac{2}{R_{\max}^2 + 1}\sum_{a^*\in\mathcal{A}}\mathbb{P}_t(x^* = a^*)\left(\mathbb{E}_t[Y_{t,a}|x^* = a^*] - \mathbb{E}_t[Y_{t,a}]\right)^2.
\tag{A.12}
$$

We bound the information ratio of IDS by the information ratio of TS:

$$\Psi_{*,2} \le \max_{t \in [n]} \frac{\langle \pi_t^{\mathrm{TS}}, \Delta_t \rangle^2}{\langle \pi_t^{\mathrm{TS}}, I_t \rangle}.$$

Using the matrix trace rank trick described in Proposition 5 in Russo and Van Roy [2014], we have $\Psi_{*,2} \le (R_{\max}^2 + 1)d/2$ in the end.

Second, we bound the information ratio with $\lambda = 3$. Recall that the exploratory policy $\mu$ is defined as

$$\max_{\mu \in \mathcal{D}(\mathcal{A})} \; \sigma_{\min}\left( \int_{x \in \mathcal{A}} xx^{\top} d\mu(x) \right).$$

Consider a mixture policy $\pi_t^{\mathrm{mix}} = (1 - \gamma)\pi_t^{\mathrm{TS}} + \gamma\mu$ where the mixture rate $\gamma \ge 0$ will be decided later. Then we will bound the following in two steps.

$$\Psi_{t,3}(\pi_t^{\mathrm{mix}}) = \frac{\langle \pi_t^{\mathrm{mix}}, \Delta_t \rangle^3}{\langle \pi_t^{\mathrm{mix}}, I_t \rangle}.$$

**Step 1: Bound the information gain**   According the lower bound of information gain in Eq. (A.12),

$$\langle \pi_t^{\mathrm{mix}}, I_t \rangle \ge \frac{2}{(R_{\max}^2 + 1)} \sum_{a \in \mathcal{A}} \pi_t^{\mathrm{mix}}(a) \sum_{a^* \in \mathcal{A}} \mathbb{P}_t(x^* = a^*) \left( \mathbb{E}_t[Y_{t,a}|x^* = a^*] - \mathbb{E}_t[Y_{t,a}] \right)^2$$

$$= \frac{2}{(R_{\max}^2 + 1)} \sum_{a \in \mathcal{A}} \pi_t^{\mathrm{mix}}(a) \sum_{a^* \in \mathcal{A}} \mathbb{P}_t(x^* = a^*) \left( a^{\top} \mathbb{E}_t[\theta^*|x^* = a^*] - a^{\top} \mathbb{E}_t[\theta^*] \right)^2.$$

By the definition of the mixture policy, we know that $\pi_t(a) \ge \gamma\mu(a)$ for any $a \in \mathcal{A}$. Then we have

$$\langle \pi_t^{\mathrm{mix}}, I_t \rangle \ge \frac{2}{(R_{\max}^2 + 1)} \gamma \sum_{a^* \in \mathcal{A}} \mathbb{P}_t(x^* = a^*)$$

$$\cdot \sum_{a \in \mathcal{A}} \mu(a)(\mathbb{E}_t[\theta^*|x^* = a^*] - \mathbb{E}_t[\theta^*])^{\top} aa^{\top} (\mathbb{E}_t[\theta^*|x^* = a^*] - \mathbb{E}_t[\theta^*]).$$

From the definition of minimum eigenvalue, we have

$$\langle \pi_t^{\mathrm{mix}}, I_t \rangle \ge \frac{2\gamma}{(R_{\max}^2 + 1)} \sum_{a \in \mathcal{A}} \mathbb{P}_t(x^* = a)C_{\min} \left\| \mathbb{E}_t[\theta^*|x^* = a^*] - \mathbb{E}_t[\theta^*] \right\|_2^2.$$

**Step 2: Bound the instant regret**   We decompose the regret by the contribution from the exploratory policy and the one from TS:

$$\langle \pi_t^{\mathrm{mix}}, \Delta_t \rangle$$

$$= \sum_a \mathbb{E}_t\left[ \langle x^*, \theta^* \rangle - \langle a, \theta^* \rangle \right] \pi_t^{\mathrm{mix}}(a),$$

$$= (1 - \gamma) \sum_a \pi_t^{\mathrm{TS}}(a)\mathbb{E}_t\left[ \langle x^*, \theta^* \rangle - \langle a, \theta^* \rangle \right] + \gamma \sum_a \mathbb{E}_t\left[ \langle x^*, \theta^* \rangle - \langle a, \theta^* \rangle \right] \mu(a) \qquad \text{(A.13)}$$

$$= (1 - \gamma) \sum_a \mathbb{P}_t(x^* = a)\mathbb{E}_t\left[ \langle x^*, \theta^* \rangle - \langle a, \theta^* \rangle \right] + \gamma \sum_a \mathbb{E}_t\left[ \langle x^*, \theta^* \rangle - \langle a, \theta^* \rangle \right] \mu(a)$$

Since $R_{\max}$ is the upper bound of maximum expected reward, the second term can be bounded $2R_{\max}\gamma$. Next we bound the first term as follows:

$$\sum_a \mathbb{P}_t(x^* = a)\mathbb{E}_t\left[ \langle x^*, \theta^* \rangle - \langle a, \theta^* \rangle \right]$$

$$= \sum_a \mathbb{P}_t(x^* = a)\left( \mathbb{E}_t[\langle a, \theta^* \rangle|x^* = a] - \mathbb{E}_t[\langle a, \theta^* \rangle] \right)$$

$$= \sum_a \mathbb{P}_t^{1/2}(x^* = a)\mathbb{P}_t^{1/2}(x^* = a)\left( \mathbb{E}_t[\langle a, \theta^* \rangle|x^* = a] - \mathbb{E}_t[\langle a, \theta^* \rangle] \right)$$

$$\le \sqrt{\sum_a \mathbb{P}_t(x^* = a)\left( \mathbb{E}_t[\langle a, \theta^* \rangle|x^* = a] - \mathbb{E}_t[\langle a, \theta^* \rangle] \right)^2},$$

where we use Cauchy-Schwarz inequality. Since all the optimal actions are sparse, any action $a$ with $\mathbb{P}_t(x^* = a) > 0$ must be sparse. Then we have

$$\left(a^\top(\mathbb{E}_t[\theta^*|x^* = a] - \mathbb{E}_t[\theta^*])\right)^2 \le s^2 \left\|\mathbb{E}_t[\theta^*|x^* = a^*] - \mathbb{E}_t[\theta^*]\right\|_2^2,$$

for any action $a$ with $\mathbb{P}_t(x^* = a) > 0$. This further implies

$$
\begin{aligned}
&\sum_a \mathbb{P}_t(x^* = a)\mathbb{E}_t\Big[\langle x^*, \theta^*\rangle - \langle a, \theta^*\rangle\Big] \\
&\le \sqrt{\sum_a \mathbb{P}_t(x^* = a)s^2 \left\|\mathbb{E}_t[\theta^*|x^* = a^*] - \mathbb{E}_t[\theta^*]\right\|_2^2} \\
&= \sqrt{\frac{s^2(R_{\max}^2 + 1)}{2\gamma C_{\min}} \frac{2\gamma}{(R_{\max}^2 + 1)} \sum_a \mathbb{P}_t(x^* = a)C_{\min} \left\|\mathbb{E}_t[\theta^*|x^* = a^*] - \mathbb{E}_t[\theta^*]\right\|_2^2} \\
&\le \sqrt{\frac{s^2(R_{\max}^2 + 1)}{2\gamma C_{\min}} \langle \pi_t^{\mathrm{mix}}, I_t\rangle}.
\end{aligned}
$$
(A.14)

Putting Eq. (A.13) and (A.14) together, we have

$$\langle \pi_t^{\mathrm{mix}}, \Delta_t\rangle \le \sqrt{\frac{s^2(R_{\max}^2 + 1)}{2\gamma C_{\min}} \langle \pi_t^{\mathrm{mix}}, I_t\rangle} + 2R_{\max}\gamma.$$

By optimizing the mixture rate $\gamma$, we have

$$\frac{\langle \pi_t^{\mathrm{mix}}, \Delta_t\rangle^3}{\langle \pi_t^{\mathrm{mix}}, I_t\rangle} \le \frac{s^2(R_{\max}^2 + 1)}{8R_{\max}^2 C_{\min}} \le \frac{s^2}{4C_{\min}}.$$

This ends the proof.

# B   Detailed algorithms

For each $a \in \mathcal{A}$, we expand $v_t(a)$ as follows:

$$
\begin{aligned}
v_t(a) &= \mathrm{Var}_t(\mathbb{E}_t[a^\top\theta|x^*]) = \mathbb{E}_t\Big[a^\top\mathbb{E}_t[\theta|x^*] - \mathbb{E}_t\big[a^\top\mathbb{E}_t[\theta|x^*]\big]\Big]^2 \\
&= \mathbb{E}_t\Big[a^\top\mathbb{E}_t[\theta|x^*] - a^\top\mathbb{E}_t[\theta]\Big]^2 = a^\top\mathbb{E}_t[(\mathbb{E}_t[\theta|x^*] - \mathbb{E}_t[\theta])(\mathbb{E}_t[\theta|x^*] - \mathbb{E}_t[\theta])^\top]a.
\end{aligned}
$$

We denote $\mu_t = \mathbb{E}_t[\theta]$ as the posterior mean and $\mu_t^a = \mathbb{E}_t[\theta|x^* = a]$. We let $\Phi \in \mathbb{R}^{|\mathcal{A}|\times d}$ as the feature matrix where each row of $\Phi$ represent each action in $\mathcal{A}$. We summarize the procedure of estimating $\Delta_t, I_t$ in Algorithm 3.

---

**Algorithm 3** Approximate $\Delta_t, v_t$ based on posterior samples

1: **Input:** $M$ posterior samples $\theta^1, \ldots, \theta^M$ from Eq. (6.2), action set $\mathcal{A}$.
2: Calculate $\widehat{\mu}_t = \sum_m \theta^m/M$.
3: **for** $a \in \mathcal{A}$ **do**
4:     Find $\widehat{\Theta}_a = \{m \in [M] : (\Phi\theta^m)_a = \max_{a' \in \mathcal{A}}(\Phi\theta^m)_{a'}\}$.
5:     Calculate $\widehat{p}_a^* = |\widehat{\Theta}_a|/M$.
6:     Calculate $\widehat{\mu}_t^a = \sum_{m \in \widehat{\Theta}_a} \theta^m/|\widehat{\Theta}_a|$.
7:     Calculate

$$\widehat{v}_t(a) = a^\top\sum_a \widehat{p}_a^*(\widehat{\mu}_t^a - \widehat{\mu}_t)(\widehat{\mu}_t^a - \widehat{\mu}_t)^\top a, \quad \widehat{\Delta}_t(a) = \sum_{a \in \mathcal{A}} \widehat{p}_a^* a^\top\widehat{\mu}_t^a - a^\top\widehat{\mu}_t.$$

8: **end for**
9: **Output:** $\widehat{v}_t, \widehat{\Delta}_t$.

---

# C Supporting lemmas

**Lemma C.1** (First-order optimality condition.). Suppose that $f_0$ in a convex optimization problem is differentiable. Let $\mathcal{X}$ denote the feasible set. Then $x$ is optimal if and only if $x \in \mathcal{X}$ and $\nabla f_0(x)^\top (y - x) \geq 0, \forall y \in \mathcal{X}$.

**Lemma C.2** (Data processing lemma). If $Z = g(Y)$ for a deterministic function $g$, then $I(X;Y) \geq I(X;Z)$.