# OpenReview forum: "Information Directed Sampling for Sparse Linear Bandits"
_NeurIPS.cc/2021/Conference — NeurIPS 2021 Spotlight_

### Official Review · Reviewer_rJi1 · 2021-07-13

**Rating:** 7
**Confidence:** 4

**Summary:**

The authors design and analyze an IDS-based algorithm for sparse linear bandits, showing its optimality in both the data-rich and data-poor regimes.

**Ethical Concerns:**

NIL

**Limitations And Societal Impact:**

NIL

**Main Review:**

This paper nicely complements that of Hao et al. (2020) [NeurIPS] by deriving a Bayesian regret bound also in the data-rich regime. The authors show that their bound is almost optimal (up to log factors). This is nice. A neat implementation that ensure that IDS is computational efficient is also provided. One issue that is missing is whether there's also a corresponding regret bund for this computationally efficient variant of IDS. It would be good to discuss this.

The proofs look correct.

The experiments are convincing apart from the rightmost figure of Fig. 3 (for d = 100). Even though IDS outperforms all competitors, the regrets of all the algorithms, including IDS, seem to increase linearly. This suggests that the horizon is not large enough to see it tapering off. I suggest redoing this plot.

**Time Spent Reviewing:**

1 hour

---

> ### Author Response · Authors · 2021-08-07
> **Review Response**
>
> Thanks for acknowledging our contributions. We would like to respond to comments point by point.
> 1. Thank you for your suggestion. For the computational efficient IDS, we only have samples from approximate posterior distribution rather than the exact one. This will incur  a non-negligible approximation error that may depend linearly on the time horizon. This is observed previously by analyzing approximate Thompson sampling [1] or more recently for IDS [2]. We will add a remark on that.
>
> [1]. X. Lu and B. Van Roy, Ensemble Sampling.
>
> [2]. Xiuyuan Lu, Benjamin Van Roy, Vikranth Dwaracherla, Morteza Ibrahimi, Ian Osband, Zheng Wen, Reinforcement Learning, Bit by Bit.
>
> 2. Thanks for your suggestion. We will redo the plot.

---

> > ### Comment · Reviewer_rJi1 · 2021-09-02
> > **Thanks**
> >
> > Dear authors,
> > Thanks for your response. This is a very nice paper and I would like to keep my score.
> > Regards,
> > Reviewer

---

### Official Review · Reviewer_jk2v · 2021-07-16

**Rating:** 7
**Confidence:** 4

**Summary:**

The paper considers the sparse linear bandit problem from Bayesian perspective. The authors show that IDS achieves optimal Bayesian regret bounds for both data-poor and data-rich regime. Since their algorithm cannot be implemented efficiently, they propose an algorithm desired to approximate the IDS, which is shown to work well empirically.


**Ethical Concerns:**

no concerns.

**Limitations And Societal Impact:**

No. Please add some comments on its limitations and societal impact.

**Main Review:**

[main]

Originality 5/5: new analysis and new algorithm

Quality 4/5

Clarity 4/5: mostly clear

Significance 4/5: The algorithm would work well for small scale problems only. However, revealing the good property about IDS is important.

I think the paper makes a solid contribution via the analysis of IDS for sparse parameters and proposing a computationally tractable algorithm.

Just to make sure I understand the paper correctly, am I right that Theorem 5.3 works for any prior on \Theta? Also, am I right that the model sampled from spike-and-slab prior is almost sparse but not actually sparse?

Q: why assume the arms' L-\infty norm is bounded, not L2 norm? What breaks down if we assume that the L2 norm of the arms is bounded?

The description of Algorithm 1 can be improved. Do you mean that we perform line 3-6 M times? Also, the thinning factor T is not well-explained. Also, how should I interpret the notation $\theta^{(MT)}$?

[etc]

* I would increase the resolution of the figures and also enlarge the font sizes; currently, I have to zoom in like 800% to see details.
* L253-254: "the actions are drawn..." => this can be deleted here because the authors discuss it in L274 again.
* L278-279: I am not sure if I agree with "ESTC performs better than non-sparsity-aware algorithms in the data-poor regime but perform poorly for the data-rich horizon.". The first plot in Figure 3 is more data-rich than the third plot, yet it seems to perform still better than non-sparsity-aware algorithms at time=1000.
* L489: "Eq. (??)" => missing reference.
* L239-240: is $\rho_i$ defined somewhere?

----

**after rebuttal**

I am satisfied with the rebuttal.

**Time Spent Reviewing:**

5

---

> ### Author Response · Authors · 2021-08-07
> **Review Response**
>
> Thanks for your thoughtful and detailed review. We would like to respond to comments point by point.
>
> 1. Yes, it works for any prior. And yes, spike-and-slab prior is not exactly sparse since exact sparse prior will be computationally intractable.
> 2. If we assume the $\ell_2$-norm of the action is bounded by a constant, then the $\ell_\infty$ norm is bounded as well. So our results still hold.
> 3. Description of Algorithm 1. We will make it more clear in the final version. Algorithm 1 will need to output $M$ posterior samples. Suppose we run a Markov Chain for 100 steps $(k=1,2,\ldots,100)$ and want 10 posterior samples. Setting the thinning factor $T=10$ means every 10 steps, we pick a $\theta^k$ as the required posterior sample, e.g., $\theta^{10}, \theta^{20},\ldots, \theta^{100}$. And $\theta^{MT}$ actually mean $\theta^{M\times T}$: the parameter at $k=M\times T$ iteration. We will add more explanations for this.
> 4. Experiments. Yes, we agree. We will re-run the experiments to make the horizon longer to see if ESTC is worse for longer-horizon. If not significant, we will remove that sentence. And we will also improve the plot in the final version.

---

### Official Review · Reviewer_GQuz · 2021-07-19

**Rating:** 7
**Confidence:** 4

**Summary:**


This paper analyses information-directed sampling (IDS) for the problem
of sparse linear bandits. They identify a key issue limiting
optimism-based algorithms (or UCB type methods) where, for certain sparse
instances, the regret can scale nearly linearly in the time horizon.
They demonstrate that IDS does circumvents this information obstacle,
by explicitly choosing actions that optimize for the information ratio:
i.e. a ratio between the (sub)optimality of the action chosen and the
future information gleaned from it. For IDS, they demonstrate an adaptive
result, that shows its (up to log factors) optimality in regret both
for short (data-poor) and long (data rich) time horizons. Computationally,
IDS is non-trivial to implement in the sparse setting, for which the authors
propose approach using mcmc sampling, borrowing from Bayesian sparse linear
regression literature.

**Limitations And Societal Impact:**

No foreseeable negative societal impact

**Main Review:**

I like the paper: it's contributions are overall well-demarcated from the
extensive literature on this and related problems, the paper provides a good high-level
introduction to the key obstacle and the proof technique they use. This is also a
topic that has traditionally been of significant interest to the community. I would recommend
its acceptance to the NeurIPS program.

Minor comments:

1. In table 1 (and other places in text) perhaps put \widetilde{O} to show that you are
omitting log factors.
2. Why does C_{min}(\cal A) need to be >= 1 for the 'exploratory actions' condition? Is
1 simply a convenient constant, e.g. does it suffice for it to be bounded away from 0?
3. In principle, there could exist instances where the regret should scale
as n^2/3 but where you do not have s-sparse optimal actions. For a lower bound
this does not matter since one needs only to demonstrate an example, but for
the achievability or upper bound, this matters. Can you comment on this: in particular,
can it be that even IDS is not adaptive to certain instances, where one does not
have the s-sparse optimal actions?
4. Notation: probably better to use \reals^{\cal A} rather than \reals^{|\cal A|} and leave
the mapping implicit.
5. Typos (likely not exhaustive): L.164: explore then *commit*, L.517: *Cauchy* Schwarz



**Time Spent Reviewing:**

2.5

---

> ### Author Response · Authors · 2021-08-07
> **Review Response**
>
> Thanks for acknowledging our contributions. We would like to respond to comments point by point.
>
> 1. Thanks for the suggestion. We will change to $\widetilde{O}$.
> 2. Yes, you are right. Simply a convenient constant. More precisely, $C_{\min}(\cal A)$ is a problem-dependent parameter of our regret bound.
> 3. We believe IDS should also adapt those instances and we view s-sparse optimal actions as a technical assumption so far. We hope to relax it in the future.
> 4. Thanks for the careful reading. We will revise the notations and typos according to suggestions.

---

### Official Review · Reviewer_p7Zh · 2021-07-29

**Rating:** 7
**Confidence:** 4

**Summary:**

The paper studies a high-dimensional linear bandit problem with sparse structure. The authors propose information-directed sampling (IDS), which naturally 4 balances the information-regret trade-off. They establish information-theoretic Bayesian regret bounds that nearly match existing lower bounds. For the implementation of sparse IDS, the authors propose an empirical Bayesian approach for sparse posterior sampling using a spike-and-slab Gaussian-Laplace prior. The numerical experiments show that the proposed method is more effective compared to the existing methods.

**Main Review:**

The paper is well-written and presents a theoretically sound method that tackles an important problem. It provides the first analysis using information theory for the sparse linear bandit problem. The result on the sub-optimality of optimism-based algorithms in sparse linear bandits is also nice.

The authors assume sparsity s is known but says (in Line 48) that this can be relaxed by putting a prior on it. I expect that a wrong prior on s can give a suboptimal result and the prior on s should be well-specified. I would like to ask the authors to elaborate on this.

In Lines 122-123, the authors argue that "the confidence set can be constructed through an online-to-confidence set conversion approach [Abbasi-Yadkori et al., 2012]" but can you construct the confidence set in a computationally efficient manner?

**Time Spent Reviewing:**

4

---

> ### Author Response · Authors · 2021-08-07
> **Review Response**
>
> Thanks for your thoughtful review. We would like to respond to comments point by point.
> 1. About the prior.
>
> We agree on this. In this work we consider Bayesian regret that averages over the instances from the prior distribution and we assume the algorithm uses the right prior which is kind of standard in literature [1][2]. It would be interesting to understand how the misspecification of prior distribution affects the regret. We will add a remark on that.
>
> [1]. D. Russo and B. Van Roy,  An Information-Theoretic Analysis of Thompson Sampling.
>
> [2]. D. Russo and B. Van Roy, Learning to Optimize Via Information-Directed Sampling.
>
> 2. We do not have a computationally efficient way to construct the sparse confidence set. Note that our algorithm does not use an online-to-confidence set conversion approach. Our lower bound described in Lines 122-123 further shows that even if you could construct such a confidence set, for some problem instances of sparse linear bandits, you will still suffer linear regret.

---

> > ### Comment · Reviewer_p7Zh · 2021-09-02
> > **Update**
> >
> > After reviewing the author's responses to my and other reviewers' comments, I raise my score. I think this is a good paper.

---

### Decision · Program_Chairs · 2021-09-27

**Decision:**

Accept (Spotlight)

**Comment:**

This paper studies sparse linear bandits in the Bayesian setting. Theoretically, it shows that information-directed sampling (IDS) has regret guarantees that adapt to (1) small/large action sets; (2) explorative action sets. Empirically, it gives a practical approximate implementation of IDS based on an empirical Bayesian approach.

All reviewers are in consensus that this paper makes a solid contribution to a problem of good interest to the NeurIPS community. In the final version, please incorporate the reviewers' suggestion and re-run the experiments if necessary.